# Enhanced antioxidant activity of *Chenopodium formosanum* Koidz. by lactic acid bacteria: Optimization of fermentation conditions

Hsing-Chun Kuo[1,2,3,4], Ho Ki Kwong[5], Hung-Yueh Chen[6], Hsien-Yi Hsu[7,8], Shu-Han Yu[5], Chang-Wei Hsieh[9], Hui-Wen Lin[10], Yung-Lin Chu[11]*, Kuan-Chen Cheng[5,6,10,12]*

1 Division of Basic Medical Sciences, Department of Nursing, Chang Gung University of Science and Technology, Chiayi, Taiwan, 2 Chang Gung Memorial Hospital, Chiayi, Taiwan, 3 Research Center for Industry of Human Ecology, Chang Gung University of Science and Technology, Taoyuan, Taiwan, 4 Chronic Diseases and Health Promotion Research Center, CGUST, Chiayi, Taiwan, 5 Institute of Biotechnology, College of Bioresources and Agriculture, National Taiwan University, Taipei, Taiwan, 6 Institute of Food Science and Technology, College of Bioresources and Agriculture, National Taiwan University, Taipei, Taiwan, 7 Department of Materials Science and Engineering, School of Energy and Environment, City University of Hong Kong, Kowloon Tong, Hong Kong, China, 8 Shenzhen Research Institute of City University of Hong Kong, Shenzhen, China, 9 Department of Food Science and Biotechnology, National Chung Hsing University, Taichung, Taiwan, 10 Department of Optometry, Asia University, Wufeng, Taichung, Taiwan, 11 Department of Food Science, College of Agriculture, National Pingtung University of Science and Technology, Pingtung, Taiwan, 12 Department of Medical Research, China Medical University Hospital, China Medical University, Taichung, Taiwan

* kccheng@ntu.edu.tw (KCC); ylchu@mail.npust.edu.tw (YLC)

**Data Availability Statement:** All relevant data are within the paper and its Supporting information files.

## Abstract

In this study, different probiotics commonly used to produce fermented dairy products were inoculated independently for *Chenopodium formosanum* Koidz. fermentation. The strain with the highest level of antioxidant activity was selected and the fermentation process was further optimized via response surface methodology (RSM). *Lactobacillus plantarum* BCRC 11697 was chosen because, compared to other lactic acid bacteria, it exhibits increased free radical scavenging ability and can produce more phenolic compounds, DPPH (from 72.6% to 93.2%), and ABTS (from 64.2% to 76.9%). Using RSM, we further optimize the fermentation protocol of BCRC 11697 by adjusting the initial fermentation pH, agitation speed, and temperature to reach the highest level of antioxidant activity (73.5% of DPPH and 93.8% of ABTS). The optimal protocol (pH 5.55, 104 rpm, and 24.4˚C) resulted in a significant increase in the amount of phenolic compounds as well as the DPPH and ABTS free radical scavenging ability of BCRC 11697 products. The $IC_{50}$ of the DPPH and ABTS free radical scavenging ability were 0.33 and 2.35 mg/mL, respectively, and both protease and tannase activity increased after RSM. An increase in lower molecular weight (<24 kDa) protein hydrolysates was also observed. Results indicated that djulis fermented by *L. plantarum* can be a powerful source of natural antioxidants for preventing free radical-initiated diseases.

**Funding:** This project was funded by the Ministry of Science and Technology, Taiwan (MOST 109-2628-E-002-007-MY3 and MOST 107-2320-B-255-001-MY3). HC Kuo received grant of MOST 107-2320-B-255-001-MY3. KC Cheng received grant of MOST 109-2628-E-002-007-MY3. URL: https://www.most.gov.tw/?l=en. HC Kuo also received funding grant of BMRPD42 for this study was also provided in part by research grants from the Chang Gung Memorial Hospital, Chiayi, Taiwan. The funders had no role in study design, data collection and analysis, decision to publish, or preparation of the manuscript.

**Competing interests:** The authors have declared that no competing interests exist.

## Introduction

Djulis (*Chenopodium formosanum* Koidz.) is a traditional crop from the same genus as quinoa (*Chenopodiun quinoa*), and it is cultivated and consumed as food or used as a wine starter in Taiwan [1]. Reports have shown that djulis exhibits beneficial effects on anti-inflammation, anti-diabetes, anti-oxidation, and immune regulation [1, 2]. Bioactive components and pigments such as peptides, betacyanin (red), betaxanthins (yellow), and polyphenols contribute to the aforementioned effects. Other ingredients, such as rutin and chlorogenic acid, can also restore the injury from UVB on HaCaT cells by reducing the level of interleukin-6 and reactive oxygen species (ROS) [3].

Lactic acid bacteria (LAB) are widely known strains of probiotics. Several studies have indicated that LAB exhibit multiple functions, such as modulating gut health, improving liver function, and decreasing cholesterol levels and blood pressure [4–6]. LAB can also enahnce the flavor of fermented products and increase the amount of antioxidative compounds in dairy products through bioconversion [7]. For example, Hsieh et al. [8] reported that heat killed cells and cytoplasmic fraction forms of *Lactobacillus acidophilus* BCRC 14079 grown in taro waste medium showed enhanced anti-tumor and immune-modulatory properties. Bianchi et al. [9] reported that synbiotic fermented beverages combining quinoa and soy had favorable nutritional, rheological, and sensory characteristics.

Traditionally, optimal fermentation condition are determined using a one-factor-at-a-time approach [10]. However, this method is both time-consuming and costly in terms of materials and human resources. In worst-case scenarios, the interactions among parameters are often overlooked, resulting in misleading conclusions. As an alternative, response surface methodology (RSM) is a statistical method for simultaneously validating the effects of and interactions among different parameters [8, 11]. RSM has been used in various fermentation applications such as wine making [12], bioethanol production [13], exopolysaccharides [14], and biomass production [8, 15].

The aim of the present study was to select suitable LAB strains for djulis fermentation to enhance antioxidant activity for the development of health-promoting beverages. Antioxidant activity of LAB-fermented djulis was evaluated according to 2,2'-azinobis-(3-ethylbenzthiazoline-6-sulphonate) (ABTS) and 2,2-diphynyl-1-picrylhydrazyl (DPPH) free radical scavenging ability. We employed RSM to determine the optimal fermentation parameters (initial pH, agitation speed and cultivation temperature) for achieving the highest level of antioxidant activity. Possible causes for the increased bioactivity after djulis fermentation and composition analysis of djulis samples before and after fermentation were also investigated.

## Materials and methods

### Materials

Domestic djulis was purchased from Pingtung, Taiwan. Lactobacilli MRS Broth was provided by Hardy diagnostics (Santa Maria, CA, USA). We purchased 95% Ethanol and methanol from Echo Chemical, Co., Ltd. (Taipei, Taiwan), and 2,2'-Azino-bis(3-ethylbenzothiazoline-6-sulfonic acid) diammonium salt (ABTS), gallic acid, Folin–Ciocâlteu phenol reagent, and α, α-diphenyl-β-picrylhydrazyl (DPPH) from Sigma-Aldrich Co. (St. Louis, MO, USA). Agar and peptone were provided by Bioshop Inc. (Burlington, ON, Canada).

### Microorganisms and medium

*Bifidobacterium infantis* BCRC14602, *Bifidobacterium adolescentis* BCRC14606, *Bifidobacterium bifidum* BCRC14615, *Bifidobacterium longum* BCRC14634, *Bifidobacterium breve*

BCRC11846, *Lactobacillus rhamnosus* GG BCRC16000, *Lactobacillus delbrueckii* subsp. *bulgaricus* BCRC10696, *Lactobacillus plantarum* BCRC11697, *Lactobacillus acidophilus* BCRC14079, *Streptococcus salivarius* subsp. *thermophiles* BCRC14085 were purchased from Bioresource Collection and Research Center (BCRC, Hsinchu city, Taiwan). All LAB strains were grown in MRS medium (Sigma-Aldrich, MI, USA). For storage, stock cultures were kept in 20% glycerol at -80˚C. Viable cells were grown in MRS medium at 37˚C for 20 hours as inoculum and sub-cultured twice a month [16]. The standard growth curve was measured at 600 nm using a Multiskan GO microplate spectrophotometer (Thermo Scientific, Waltham, MA, USA).

## Djulis fermentation

Djulis was crushed into powder, filtered using a 0.6 mm mesh and then stored at -20˚C until use. Djulis powder was combined with 10 times the amount of $ddH_2O$ (w/v) and sterilized at 90˚C for 10 minutes. After cooling to room temperature, samples were inoculated with 1% LAB (~7 log CFU mL $^{-1}$) as a seed culture and then fermented at 37˚C for 48 hours. To determine the optimal fermentation time, LAB samples were taken every 12 hours and monitored for their pH, bacteria number, total phenolic content (TPC), ABTS, and DPPH. The optimal LAB was chosen based on the ABTS and DPPH assay results.

## Anti-oxidant activity

**DPPH assay.** Fermented djulis samples were freeze-dried and diluted to 5 mg/mL (deionized water). They were then mixed with DPPH ethanol solution (100 μM) at 1:1 ratio in a 96-well tissue culture plate to carry out the reaction in the dark for 30 minutes. Finally, the samples were analyzed at the 517 nm wavelength using the microplate spectrophotometer [17]. DPPH scavenging activity of the djulis extracts was calculated as follows:

$$DPPH\ scavenging\ activity(\%) = [1 - (A1 - A2)/A0] \times 100. \tag{1}$$

where A0 = DPPH (without samples), A1 = Sample + DPPH, and A2 = Sample (without DPPH). We used this method to screen the optimal LAB for djulis fermentation. Once the optimal fermentation protocol was evaluated, the IC50 was calculated to provide an absolute number. $IC_{50}$ of the djulis samples was obtained from the regression curve between concentration and DPPH scavenging activity.

**ABST assay.** Fermented djulis samples were freeze-dried and diluted to 5 mg/mL (80% methanol). A 2.5 mM $K_2O_8S_2$ solution was prepared with $K_2O_8S_2$ and 7 mM 2,2'-azinobis (3-ethylbenzothia-zoline-6-sulfonic acid) solution. The solution was placed in the dark for 12−16 hours until it became blue-green due to the formation of $ABTS^{.+}$. The $ABTS^{.+}$ solution was diluted with 0.2 M phosphate buffer solution (pH 7.4) until its $OD_{734}$ reached to 0.7±0.02. Each sample (3 μL) was added to $ABTS^{.+}$ solution (300 μL) in the dark for 6 minutes [17]. After the reaction was completed, the samples were analyzed at 734 nm using the microplate spectrophotometer. The ABTS scavenging activity of the djulis was calculated as follows:

$$ABTS(\%) = [AC - AS]/AC \times 100. \tag{2}$$

where AC = ABTS (without samples); AS = Sample + ABTS.

**Determination of the TPC.** The TPCs was determined using Folin–Ciocâlteu's reagent following the methods described by Wu et al. [13] but with slight modifications. Briefly, each extract (100 mg) was dissolved in a solution of 5 mL of 3% HCl in methanol/deionized water (1:1), and the resulting mixture (100 μL) was added to 100 μL of 10% aqueous sodium carbonate solution. After 2 minutes, 100 μL of 50% Folin–Ciocâlteu's reagent was added to the

mixture. After the solution had stood for 30 minutes, absorbance was measured at 750 nm against a blank. TPC was calculated based on the calibration curve of gallic acid, and this is reported as mg gallic acid equivalent per 1 g of dry djulis powder (mg GAE/1 gdw).

**Enzyme activity.** Protease activity was determined by taking aliquots of 100 μL of the fermented liquid and adding 100 μL of 0.1 M sodium phosphate buffer (pH 5.7). To this mixture, 100 μL of substrate was added and incubated for 30 minutes at 50˚C for the two cultivars. The reaction was stopped by adding 500 μL of trichloroacetic acid at 10% (v/v) and centrifuged at 10,000 x g for 5 minutes. We then added 200μL of 1.8 M NaOH to the supernatant. Readings were taken using a spectrophotometer at 280nm. For quantification, an enzymatic unit was considered the amount of enzyme required to increase the absorbance by 0.01 [18]. To measure the tannase activity, the sample solution (100 μL) was incubated with 300 μL of 1.0% (w/v) tannic acid within a 0.2 M acetate buffer (pH 5.0) at 40˚C for 30 minutes. The reaction was then terminated at 0˚C by adding 2 mL bovine serum albumin (1 mg/mL), causing the remaining tannic acid to precipitate out of the solution. The samples were then centrifuged (5,000 x g, 10 min), and the precipitate was dissolved in 2 mL of sodium dodecyl sulfate (SDS)–triethanolamine (1% w/v, triethanolamine) solution. Absorbency was measured at 550 nm after addition of 1 mL of $FeCl_3$ (0.13 M). One unit of tannase was defined as the amount of enzymerequired to hydrolyze 1μ mole of ester linkage of tannic acid in 1 minute under specific conditions [19].

**Optimization for djulis fermentation.** RSM using the Box-Behnken design was performed to select the optimal conditions for djulis fermentation. Three variables, namely the initial pH (5, 6, 7), agitation speed (50, 100, 150 rpm), and cultivation temperature (20, 25, 30˚C), were optimized based on the results of a set of experiments. A total of 15 runs were performed to establish a model and predict the optimal conditions. Three levels of design were introduced: low, medium, and high (respectively labeled as -1, 0, and 1 in Table 1. Minitab software was used to predict the optimal values of the three variables according to the following second-order polynomial equation:

$$Y = B_0 + \sum B_i X_i + \sum B_{ii} X_i^2 + \sum \sum B_{ij} X_i X_j$$

where $Y$ is the dependent variable and represents the predicted response on ABTS ability; $B_0$ represents the fitted response at the design's center point; $B_i$, $B_{ii}$, and $B_{ij}$ are the coefficient for linear, quadratic, and cross-product regression, respectively; and $X_i$ and $X_j$ (with $j = i + 1$) are the coded independent variables ($X_1$ = initial pH, $X_2$ = agitation speed, and $X_3$ = fermentation temperature).

**SDS gel electrophoresis.** Djulis samples weighing approximately 0.1 g were mixed with 15 mL of deionized water and stirred for 30 minutes at room temperature. Next, 0.1 M NaOH was added to adjust the pH to 9.0, and the samples were stirred for another 30 minutes, centrifuged at 4,500 × g for 20 minutes, and then 0.1 N HCL was added to adjust the pH to 5.0. After removal of the supernatant, the samples were mixed with 500 μL of 63 mM Tris-HCl solution (pH 8.0). The obtained djulis protein samples were quantitated with a protein assay kit (Bio-

**Table 1. Range and corresponding levels of the independent variables.**

| Variable | Range values of coded levels | | |
|---|---|---|---|
| | **−1** | **0** | **1** |
| pH ($X_1$) | 5 | 6 | 7 |
| Rpm ($X_2$) | 50 | 100 | 150 |
| Temperature ($X_3$, ˚C) | 20 | 25 | 30 |

Rad Laboratories, Hercules, CA, USA) [20]. The total proteins were used for SDS gel electrophoresis analysis, and all samples were subjected to 15% SDS-polyacrylamide gel electrophoresis (SDS–PAGE) for 180 minutes at 60 V (stacking gel) and 120 V (separating gel). The separating gel was shaken and washed with deionized water three times at 70 rpm. After the deionized water was removed, RAPIDStain was added to completely submerge the gel, and it was shaken at 70 rpm for 1 hour. Finally, the gel was washed with two to three times deionized water at 70 rpm (10–15 minutes/time).

**Composition analysis of fermented djulis samples.** Fermented djulis samples were analyzed for their moisture, carbohydrate, protein, fat, and ash content following methods described in the literature [13] but with slight modifications. Crude protein content (g/100 g dry matter) was evaluated using the Kjeldahl method on the basis of nitrogen level and multiplied by 5.7. Crude fat content (g/100 g dry matter) was measured using Soxhlet extraction with petroleum ether. Moisture was determined by oven-drying at 105°C. Ash content was determined by placing samples overnight in a furnace at 600°C. Total carbohydrate content (g/100 g dry matter) was obtained by taking the difference between 100 and the sum of the ash content, moisture, crude fat, and crude protein.

**Statistical analysis.** All experiments were conducted with three independent evaluations and with three replications for each sample. Values are expressed as the mean ± SD. Minitab software (Minitab Inc., University City, Pennsylvania, USA) was used to perform one-way ANOVA and Duncan's new multiple range tests as well as RSM evaluation and analysis. Differences were considered statistically significant differences where $p < 0.05$.

# Results and discussions

## Strain selection for djulis fermentation

To improve the functional properties, nutritional value, and taste of djulis, we fermented djulis grains using different strains of LAB. Djulis samples were inoculated with 10 strains of LAB, and were measured for their DPPH and ABTS radical scavenging activity after 24 hours of fermentation. The results showed that all 10 strains of LAB promoted the antioxidant activity of djulis. However, amongst all samples, *L. plantarum* BCRC 16000 and 11697 exhibited relatively higher performance for DPPH activity (93.2%) (Fig 1A). Additionally, the ABTS radical scavenging activity assay demonstrated that *L. plantarum* BCRC 11697 showed significantly higher ABTS radical scavenging activity (76.9%) compared to the control (64.2%) ($p < 0.05$). Fig 1C shows the TPC was in line with the results regarding *L. plantarum* BCRC 11697 fermentation and antioxidant activity. *L. plantarum* BCRC 11697 produced more phenolic compounds than did the other strains. Past studies have reported that a correlation coefficient of R = 0.966 between ABTS and TPC, and R = 0.939 between DPPH and TPC [21]. Turkan [22] also reported that polyphenols are antioxidants that reduce ROS and reactive nitrogen species. This could explain why *L. plantarum* BCRC 11697 performed the best in the DPPH and ABTS tests. *L. plantarum* is a common LAB strain used in fermented plant-based foods, and it is often applied to the metabolic bacteria model of phenolic compounds [23]. Moreover, it can degrade phenolic compounds in food, and produce some compounds that affect food flavor and enhance antioxidant activity [23]. For example, all strains of *L. plantarum* secretes TanBLp (tannase), and tannase can hydrolyze the ester bond of gallic acids and protocatechuic acids [23]. Furthermore, feruloyl esterases are involved in releasing enzymes from plant cell walls and promoting antioxidant activity [24]. ABTS and DPPH assays are widely used methods for assessing antioxidant activity in natural herbal products. Both assays are spectrophotometric techniques based on the quenching of stable colored radicals (ABTS or DPPH) and show the radical scavenging ability of antioxidants even when present in complex plant extracts [25].

(A)

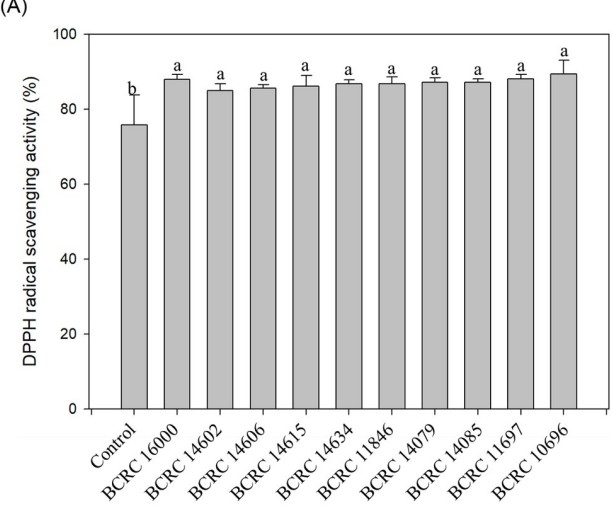

(B)

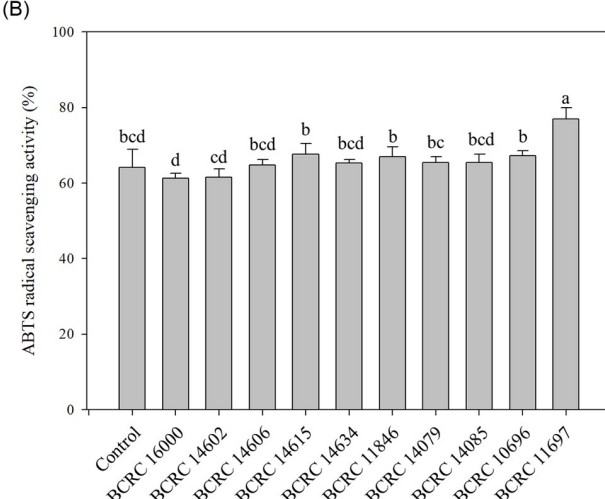

(C)

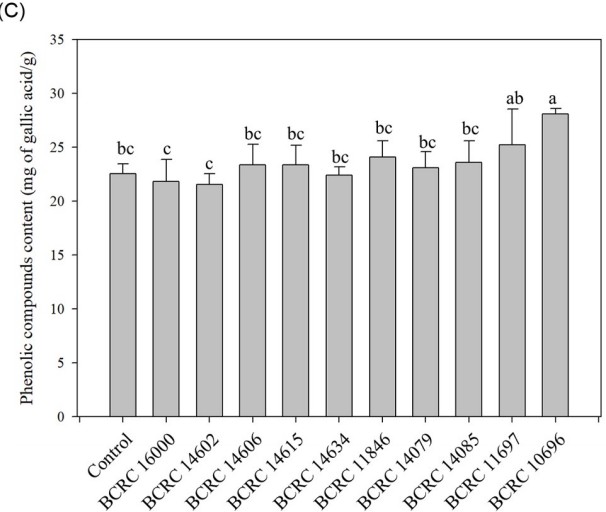

**Fig 1.**

Both methods are rapid, simple, inexpensive and widely used to measure the ability of compounds to act as free radical scavengers or hydrogen donors, and to evaluate the antioxidant activity of complex extracts. Several studies have also adopted ABTS and DPPH assays to evaluate the quinoa antioxidant activity [17, 26, 27].

To determine the optimal djulis fermentation time for *L. plantarum* BCRC 11697, we carried out 54-hour cultivation with samples collected every 6 hours. As shown in Fig 1(A), all LAB strains exhibited similar DPPH scavenging effect, while BCRC11697 showed significantly higher ABTS amongst all the strains (Fig 1(B)); as such, we chose BCRC 11697 for the remainder of the study and used ABTS as the indicator.

Fig 2 shows the results of the ABTS radical scavenging activity, CFU, and pH values during fermentation. ABTS activity in the fermented djulis samples peaked after 24 hours of fermentation (77%) and then decreased gradually after 48 hours. Therefore, 24 hours was selected as the fermentation time for the remainder of the study. Moreover, lactic acidification improved the extraction of total phenols when the selected strain was used, and this has also been

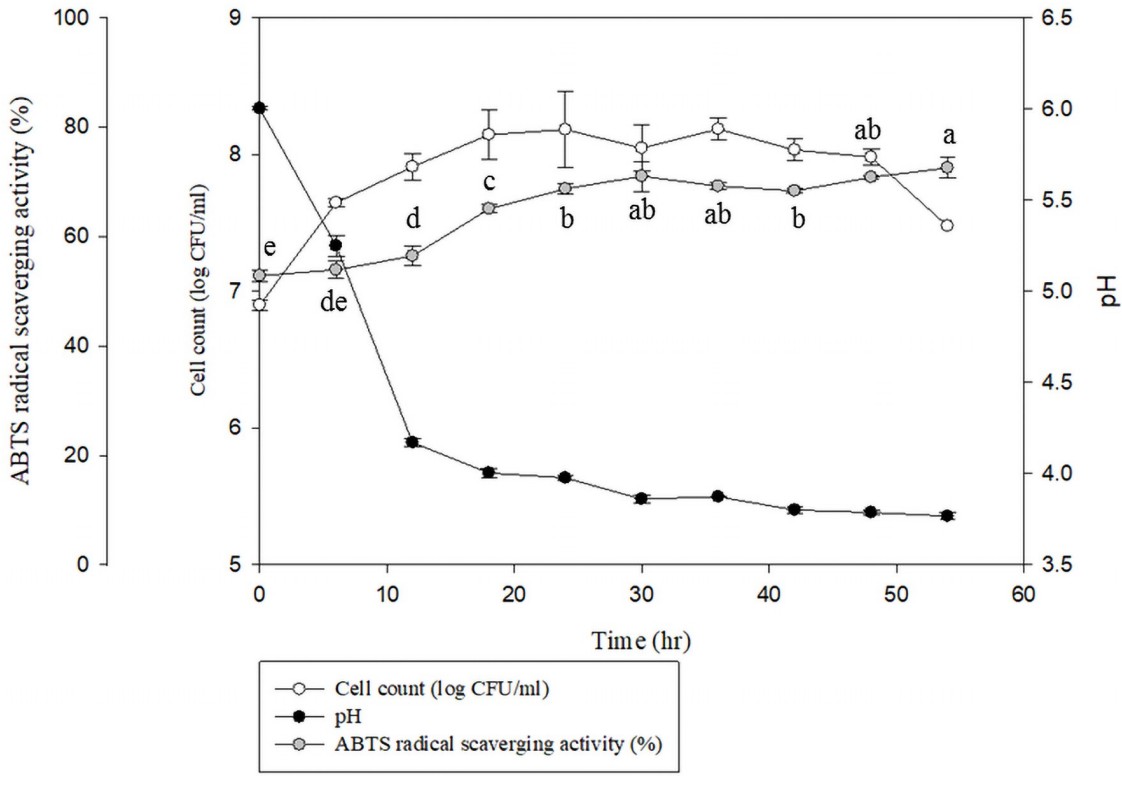

**Fig 2.**

reported in previous research [26]. Esterase activity released the complex glycosylated-phenolic compounds into the corresponding phenolic acids during fermentation.

## RSM optimization

A set of individual experiments was conducted for each variable ($X_1$: initial pH, $X_2$: agitation speed, and $X_3$: temperature) in order to elucidate the specific effect of each parameter on ABTS antioxidation ability. We investigated initial pH values ranging from 2 to 10 for solutions of djulis powder mixed with deionized water for cultivation. An initial pH of 6 yielded the highest ABTS antioxidant activity ($IC_{50}$: 2.59 mg/mL) among the fermented samples (S1A Fig). Initial pH values of 5, 6, and 7 were chosen as experimental values. Different agitation speeds were also evaluated for ABTS antioxidant ability. Results showed that samples obtained at 100 rpm of agitation had the best ABTS ($IC_{50}$: 2.39 mg/mL). Therefore, agitation speeds of 50, 100 and 150 rpm were chosen as experimental values (S1B Fig). For the temperature parameter, we evaluated temperatures ranging from 15˚C to 40˚C. Results showed the best fermentation temperature to be 25˚C, which gave the lowest ABTS $IC_{50}$: 2.44 mg/mL). Therefore, fermentation temperatures of 20˚C, 25˚C, and 30˚C were chosen as experimental values (S1C Fig). Based on the results from the one-factor-at-a-time approach, RSM was applied to determine and to optimize the three fermentation parameters in order to achieve the highest ABTS anti-oxidation ability in the fermented djulis product (Table 1). The results are summarized in Table 2. Multiple regression was applied to the experimentally determined data in Eq (1) to estimate the regression coefficients, and the following second-order polynomial equation was

**Table 2. Experimental range and values in the central composite design for optimizing the fermentation conditions.**

| StdOrder | RunOrder | PtType | Blocks | Initial pH | RPM | Tm. (˚C) | ABTS-$IC_{50}$ (mg/mL) |
|---|---|---|---|---|---|---|---|
| 12 | 1 | 2 | 1 | 6 | 150 | 30 | 3.50 |
| 8 | 2 | 2 | 1 | 7 | 100 | 30 | 3.65 |
| 5 | 3 | 2 | 1 | 5 | 100 | 20 | 3.26 |
| 9 | 4 | 2 | 1 | 6 | 50 | 20 | 3.50 |
| 7 | 5 | 2 | 1 | 5 | 100 | 30 | 3.57 |
| 4 | 6 | 2 | 1 | 7 | 150 | 25 | 3.20 |
| 2 | 7 | 2 | 1 | 7 | 50 | 25 | 3.41 |
| 11 | 8 | 2 | 1 | 6 | 50 | 30 | 4.04 |
| 14 | 9 | 0 | 1 | 6 | 100 | 25 | 2.97 |
| 3 | 10 | 2 | 1 | 5 | 150 | 25 | 3.24 |
| 1 | 11 | 2 | 1 | 5 | 50 | 25 | 3.28 |
| 10 | 12 | 2 | 1 | 6 | 150 | 20 | 3.66 |
| 6 | 13 | 2 | 1 | 7 | 100 | 20 | 3.65 |
| 15 | 14 | 0 | 1 | 6 | 100 | 25 | 3.14 |
| 13 | 15 | 0 | 1 | 6 | 100 | 25 | 2.83 |

obtained using Minitab software:

$$Y = 13.2626 - 0.4305X_1 + 0.0031X_2 - 0.7593X_3 + 0.0804X_1^2 + 0.00009X_2^2 + 0.0188X_3^2 - 0.00085X_1X_2 - 0.0152X_1X_3 - 0.0007X_2X_3.$$

where Y = ABTS free radical scavenging ability-$IC_{50}$; theory value: $X_1$ = initial pH; $X_2$ = agitation speed (rpm); $X_3$ = fermentative temperature (˚C).

The predicted optimal parameters of $X_1$, $X_2$, and $X_3$ were obtained by applying the regression analysis of Eq (2); these were pH 5.55, 104 rpm, and 24.4˚C. The predicted value of ABTS-$IC_{50}$ was 2.42 mg/mL which approximates our experimental result (2.35 mg/mL).

The coefficient of determination of the regression for the response related to the significant effects in the model was $R^2$ = 0.946 (Table 3). Hence, the sample variation of 94.6% for ABTS-$IC_{50}$ was associated with the three independent variables. The interaction between temperature and agitation speed can be observed from the results ($p < 0.05$). We hypothesize that is because the heating process can be accelerated by the increased agitation speed, resulting in a favored environment for LAB growth and TPC production. Similar results were reported by Dinarvand et al. [28]. They reported that the interaction between temperature and agitation speed affected the production of invertase from *Aspergillus niger*. The surface plots for ABTS-$IC_{50}$ are shown in Fig 3. The initial ABTS-$IC_{50}$ increased with the initial pH, reaching an optimal ABTS-$IC_{50}$ value approximately 5.55, which declined gradually above the optimal pH due to inactivation of the tannase, which accords with previous reports [23, 25]. The adequacy of the full quadratic model of liquefaction was also evaluated via ANOVA. The model summary statistics in Table 3 indicate the adequacy of the models including linear, 2-factor interactions and quadratic terms ($P < 0.05$). The lack-of-fit error was nonsignificant ($p = 0.842$), verifying the accuracy fit of the second-order model (Eq 2) to the true response of ABTS-$IC_{50}$.

For the following assay, we chose pH 5.55, 104 rpm and 24.4˚C as our fermentation conditions. Between the unfermented and fermented djulis samples, the fermented sample exhibited markedly higher antioxidant ability. For example, the $IC_{50}$ of ABTS was 3.4 mg/mL before fermentation, and this decreased to 2.35 mg/ml after optimization. In addition, the $IC_{50}$ of DPPH

**Table 3. Estimated regression coefficients for ABTS free radical scavenging ability-IC$_{50}$.**

| Sources | DF | Sum of squares | Mean squares | F-value | P-value | |
|---|---|---|---|---|---|---|
| Model | 9 | 1.24941 | 0.138824 | 9.74 | 0.011 | Significant |
| pH | 1 | 0.0386 | 0.038597 | 2.71 | 0.161 | |
| RPM | 1 | 0.04905 | 0.049049 | 3.44 | 0.123 | |
| Tm. | 1 | 0.06107 | 0.061074 | 4.29 | 0.093 | |
| pH$^2$ | 1 | 0.00356 | 0.023883 | 1.68 | 0.252 | |
| RPM$^2$ | 1 | 0.13241 | 0.186733 | 13.1 | 0.015 | Significant |
| Tm.$^2$ | 1 | 0.812 | 0.812001 | 56.98 | 0.001 | Significant |
| pH*RPM | 1 | 0.00728 | 0.007285 | 0.51 | 0.507 | |
| pH*Tm. | 1 | 0.02307 | 0.023075 | 1.62 | 0.259 | |
| RPM*Tm. | 1 | 0.12236 | 0.12236 | 8.59 | 0.033 | Significant |
| Residual error | 5 | 0.07126 | 0.014251 | | | |
| Lack of fit | 3 | 0.02082 | 0.006939 | 0.28 | 0.842 | |
| Pure error | 2 | 0.05044 | 0.025219 | | | |
| | | | | | R$^2$ | 94.60% |
| | | | | | Adjusted R$^2$ | 84.89% |

DF refers to degrees of freedom, which differs significantly (p < 0.05). The optimal starting reaction conditions for anti-oxidation were pH 5.55, 104 rpm and 24.4˚C.

was 1.11 mg/mL before fermentation, decreasing to 0.33 mg/mL (p < 0.05) after RSM (Table 4). Compared to the TPC in the unfermented djulis samples, that in the fermented samples exhibited a significant increased from 9.33 to 28.97 (mg of gallic acid/gdw) (p < 0.05). (Table 4).

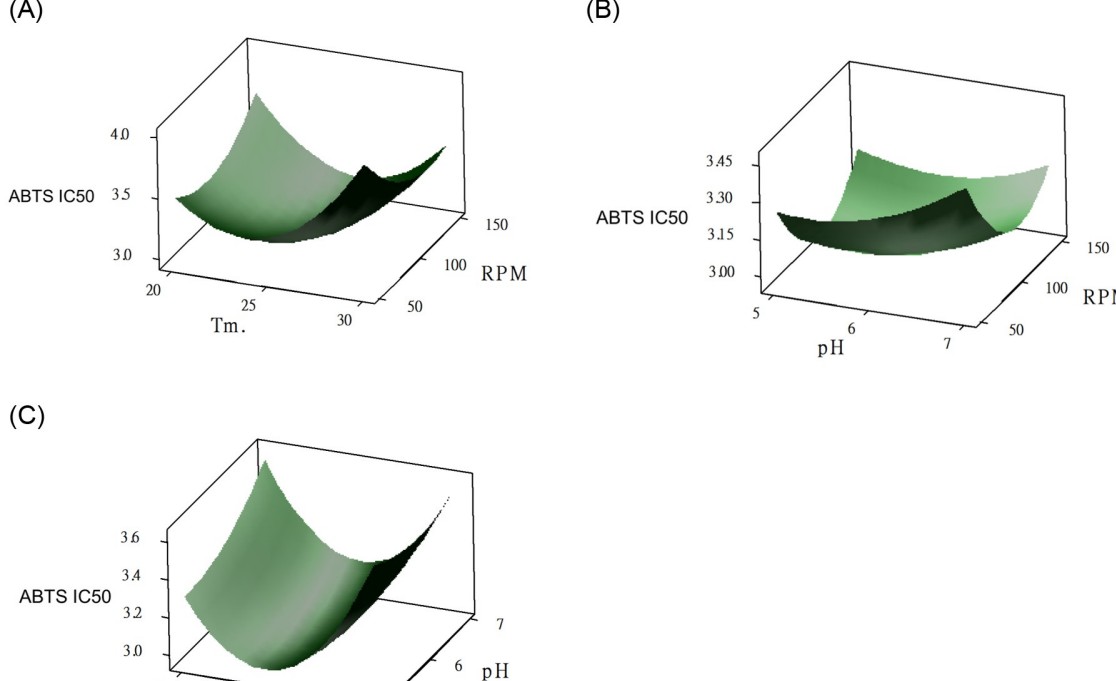

**Fig 3.**

**Table 4. Changes in TPC, cell number, pH, and DPPH and ABTS radical scavenging activity in djulis fermented with *L. plantarum* BCRC 11697.**

| Group | TCP (mg of gallic acid/gdw) | ABTS-IC$_{50}$ (mg/ml) | DPPH-IC$_{50}$ (mg/ml) | log CFU/ml | pH | Protease activity (U/mg-protein) | Tannase activity (U/mg-protein) |
|---|---|---|---|---|---|---|---|
| Unfermented | 9.33 ± 0.25 [a] | 3.40 ± 0.32[a] | 1.11 ± 0.20[a] | 7.31 ± 0.04 [a] | 5.55 ± 0.02 [a] | ND | ND |
| Fermented | 27.68 ± 0.21 [b] | 2.44 ± 0.26[b] | 1.01 ± 0.17[a] | 7.42 ± 0.08 [a] | 4.21 ± 0.03 [b] | 1.69 ± 0.17 [a] | 0.88 ±0.11 [a] |
| Fermented-RSM | 28.97 ± 0.19 [b] | 2.35 ± 0.46[b] | 0.33 ± 0.02[b] | 8.61 ± 0.32 [b] | 4.09 ± 0.01 [b] | 2.53 ± 0.21 [b] | 1.12 ± 0.09 [b] |

Statistical differences were calculated using Duncan's new multiple range test. Values are presented as the mean ± SD of three independent experiments with the different superscripts (a, b) indicating significantly differences ($p < 0.05$). ND: not detected.

The increase of TPC is due to the presence of protease (2.53 U/mg-protein), tannase (1.12 U/mg-protein), and other enzymes used in enzymatic hydrolysis. Solid-state fermentation has also been adopted for cereal grains fermentation using fungi in previous research [29] that reported significantly greater antioxidant properties in the fermented products than unfermented grains. In the case of quinoa fermentation, that study reported an increase of 2.46 mg/g in TPC content after 35 days of fermentation. For our case, the TPC content of fermented djulis increased by 19.64 mg/g after 24 hours, which provided a fast and economically feasible method for the up-scaled production of antioxidant-rich ingredients.

## Changes of djulis components through fermentation

Principal components analysis showed that the freeze-dried powder of the fermented djulis contained 68.80% carbohydrates, 17.01% crude protein, 4.12% crude fat, 5.71% ash, and 4.37% moisture (Table 5). A slight decrease in carbohydrates, protein, and fat was observed due to the presence of enzymes partaking in hydrolysis and oxidation, which was discussed in a previous study [29]. LAB utilized the carbohydrates, protein and fat of the djulis since it was the only nutrient within the medium. Some nutrients could be hydrolyzed into small molecules such as peptides, oligosaccharides, and short-chain fatty acids due to the presence of related enzymes.

## Protein hydrolyzation

Fermented grain products are consumed in many countries and are one of the most crucial sources of bioactive peptides [29]. Grains fermented using bacteria (LAB and *Bacillus* spp.) yield many different types of fermented products that possess a multitudinous array therapeutic properties, such as antioxidant, antihypertensive, antimicrobial, antidiabetic, and anticancer activity [29].

A previous study identified five peptides with antioxidant activity (ABTS and DPPH) after LAB fermentation through the hydrolysis of quinoa protein [26]. The size of each peptide was approximately 5–9 amino acid residues. Another study demonstrated that the bands between

**Table 5. Carbohydrates, protein, fat, ash and moisture of *Chenopodium formosanum* Koidz. fermented product.**

| | Carbohydrates (g/100 g) | Protein (g/100 g) | Fat (g/100 g) | Ash (g/100 g) | Moisture (g/100 g) |
|---|---|---|---|---|---|
| Quinoa | 74 | 16.3 | 7 | 2.7 | 0 |
| Djulis | 70.62 ± 0.15 | 19.15 ± 0.32 | 4.34 ± 0.17 | 2.62 ± 0.08 | 3.27 ± 0.21 |
| Dry powder of fermented djulis | 68.80 ± 0.16 | 17.01 ± 0.29 | 4.12 ± 0.54 | 5.71 ± 0.08 | 4.37 ± 0.21 |

Values are presented as mean ± SD of three independent experiments.

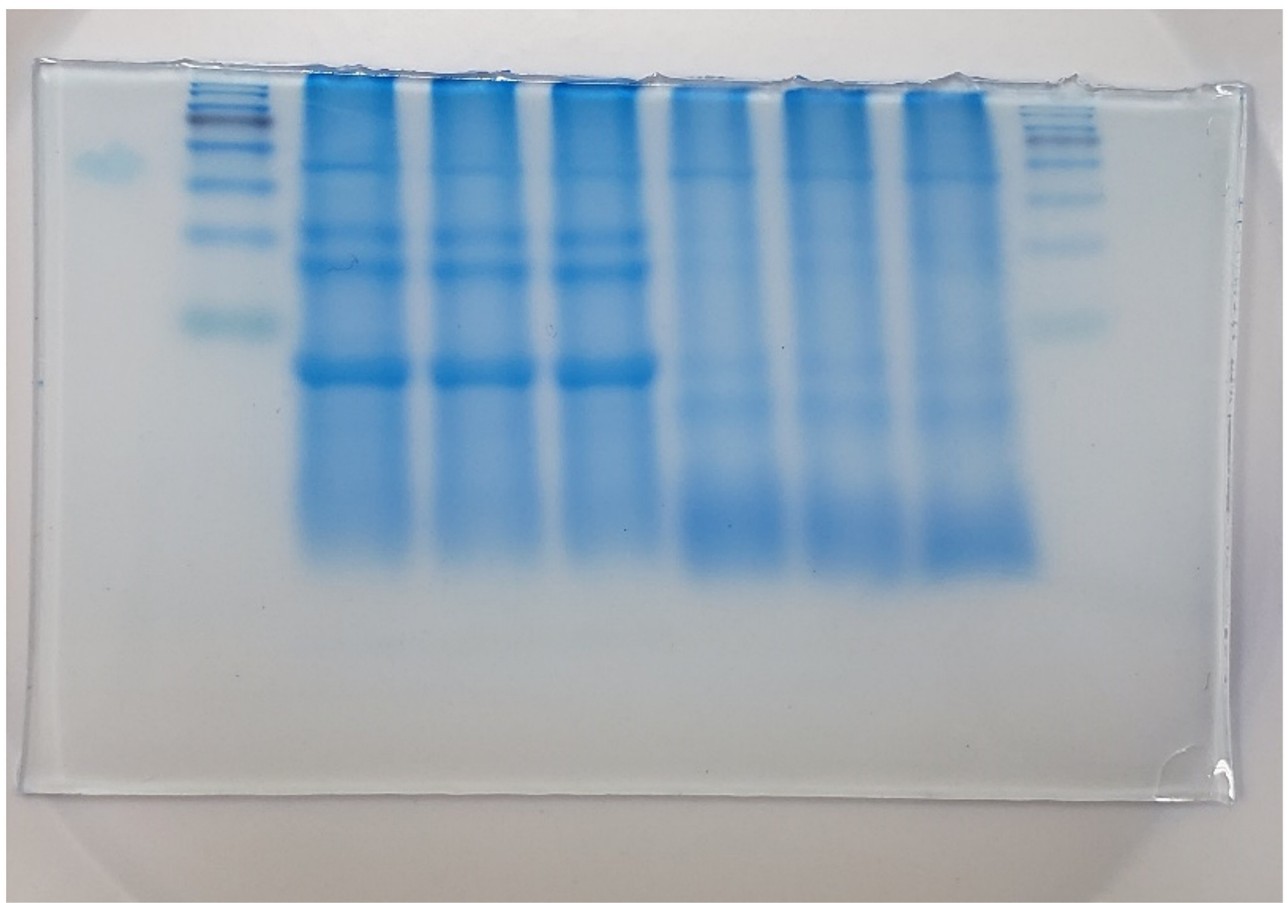

**Fig 4.**

53 and 41 kDa, two bands around 32 to 24 kDa, and bands under 24 kDa potentially have antioxidant activity as well [30]. The present study employed SDS-PAGE to determine the protein distribution of unfermented and fermented djulis. Our results showed that proteins of djulis had been hydrolyzed after fermentation. For example, two bands were between 24 and 32 kDa, one major band under 24 kDa disappeared after fermentation, and more bands with lower molecular weight were observed after fermentation (Fig 4). These results aligned with the behavior of wheat after *L. plantarum* M616 fermentation [30]. Virtanen et al. [30] also reported that milk whey protein hydrolysate weighing 4–20 kDa showed remarkably high antioxidant activity compared to that of the original milk whey protein.

## Conclusions

We applied RSM to determine the optimal fermentation conditions and to evaluate the interaction among the initial pH, agitation speed, and temperature. Our results show that *L. plantarum* BCRC 11697 is the optimal LAB strain for djulis submerged fermentation among the 10 candidates of LAB strains were investigated. After RSM-assisted optimization, we observed significant improvement in the free radical scavenging activity of DPPH and ABTS and in TPC. The presence of protease and tannase activity also supports that *L. plantarum* BCRC 11697 enhances free radical scavenging bioactivity through protein hydrolysis and the release of bound-phenolic compounds. In conclusion, fermented djulis using LAB shows potential for

commercialization as a beverage. Future studies will need to investigate the up-scaled production of djulis content, determination of the specific mechanisms of antioxidation and identification of bioactive peptides for the findings of this study to be employed in commercial applications.

## Supporting information

**S1 Fig.** (A) Initial pH-The ABTS radical scavenging activity of Chenopodium formosanum Koidz. fermented with Lactobacillus plantarum BCRC 11697. (B) RPM-The ABTS radical scavenging activity of Chenopodium formosanum Koidz. fermented with Lactobacillus plantarum BCRC 11697. (C) Fermentation temperature-The ABTS radical scavenging activity of Chenopodium formosanum Koidz. fermented with Lactobacillus plantarum BCRC 11697. Statistical differences were calculated by Duncan's new multiple range test. Values are presented as mean ± SD of three independent experiments with different superscripts (a, b, c, d) are significantly different ($p < 0.05$).
(TIF)

**S1 Raw image.**
(PDF)

**S1 Graphical abstract.**
(JPG)

## Acknowledgments

The authors greatly acknowledge the article proofreading by Iris YS Wu who is a native speaker from Molecular Environmental Biology, University of California, Berkeley (Berkeley, CA, USA).

## Author Contributions

**Conceptualization:** Hsing-Chun Kuo, Hsien-Yi Hsu, Chang-Wei Hsieh, Yung-Lin Chu, Kuan-Chen Cheng.

**Data curation:** Ho Ki Kwong, Hung-Yueh Chen.

**Formal analysis:** Ho Ki Kwong.

**Investigation:** Hsien-Yi Hsu.

**Methodology:** Hung-Yueh Chen, Shu-Han Yu, Hui-Wen Lin, Yung-Lin Chu.

**Resources:** Hsing-Chun Kuo.

**Supervision:** Hsing-Chun Kuo, Hung-Yueh Chen, Hsien-Yi Hsu, Shu-Han Yu, Chang-Wei Hsieh, Kuan-Chen Cheng.

**Validation:** Chang-Wei Hsieh.

**Writing – original draft:** Yung-Lin Chu.

**Writing – review & editing:** Shu-Han Yu, Hui-Wen Lin, Kuan-Chen Cheng.

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
