## [Decision Letter · Decision Letter 0]

8 Dec 2020

PONE-D-20-34408

Enhanced Antioxidant Activity of Chenopodium formosanum K oidz. by Lactic Acid Bacteria: Optimization of Fermentation Conditions

PLOS ONE

Dear Dr. Cheng,

Thank you for submitting your manuscript to PLOS ONE. After careful consideration, we feel that it has merit but does not fully meet PLOS ONE’s publication criteria as it currently stands. Therefore, we invite you to submit a revised version of the manuscript that addresses the points raised during the review process.

Authors should pay more attention to the aims of the study (Introduction section) and how they present results in the Results section. Calculation methods of antioxidant assays may be questionable. Some statistical significance might be presented in an inappropriate manner, as noted in the Reviewers' reports. Labels in Fig 1A and Fig 1B are not visible. Can they stay in an oblique way, as in Fig 1C? Figure 3 should be prepared according to Submission Guidelines specified for figures: https://journals.plos.org/plosone/s/figures.

Language needs considerable improvement.

We look forward to receiving your revised manuscript.

Kind regards,

Branislav T. Šiler, Ph.D.

Academic Editor

PLOS ONE

Journal Requirements:

2. Please include your tables as part of your main manuscript and remove the individual files. Please note that supplementary tables should be uploaded as separate "supporting information" files.

"This project was funded by the Ministry of Science and Technology, Taiwan (MOST

 109-2628-E-002-007-MY3 and MOST 107-2320-B-255-001-MY3). The funding

grant of BMRPD42 for this study was also provided in part by research grants from

the Chang Gung Memorial Hospital, Chiayi, Taiwan."

"This project was funded by the Ministry of Science and Technology, Taiwan (MOST 109-2628-E-002-007-MY3 and MOST 107-2320-B-255-001-MY3). HC Kuo received grant of MOST 107-2320-B-255-001-MY3. KC Cheng received grant of MOST 109-2628-E-002-007-MY3. URL: https://www.most.gov.tw/?l=en . The funders had no role in study design, data collection and analysis, decision to publish, or preparation of the manuscript."

Reviewers' comments:

Reviewer's Responses to Questions

**Comments to the Author**

1. Is the manuscript technically sound, and do the data support the conclusions?

Reviewer #1: Partly

Reviewer #2: Yes

Reviewer #3: Yes

2. Has the statistical analysis been performed appropriately and rigorously? 

Reviewer #1: Yes

Reviewer #2: Yes

Reviewer #3: Yes

3. Have the authors made all data underlying the findings in their manuscript fully available?

Reviewer #1: Yes

Reviewer #2: Yes

Reviewer #3: Yes

4. Is the manuscript presented in an intelligible fashion and written in standard English?

Reviewer #1: Yes

Reviewer #2: Yes

Reviewer #3: Yes

5. Review Comments to the Author

Reviewer #1: Overall, the manuscript provides a good information of the subject matter. The experimental designs and analyses are standard and appropriate for the study. However, there are some concerns that need to be addressed by the authors.

1. The authors may wish to send the manuscript for a professional proof-read. There are some grammatical errors and inconsistency of sentences. Many of the scientific names of organisms throughout the manuscript are not in italic.

2. Introduction:

i) The sentence 'Lactic acid bacteria (LAB) are responsible for a group of famous strains of probiotics' (page 4, line 62-63) is somewhat misleading.

ii) The motivation/cause for this study is not clearly explained. The authors may wish to further explain the reason why they wanted to enhance the antioxidant activity and also why they selected LAB in the fermentation process. And perhaps explain why they selected ABTS and DPPH assays as the anti-oxidant assays for this study.

3. Materials and methods:

i) Page 7, line 143 - ABTS full name is repetitive.

ii) Page 9, line 193 - stated here that the authors used SAS software. But on page 13, line 300 - the authors stated that they used Minitab. Which statistical software did they actually use? Why did the authors used two different statistical software?

4. Results and discussion: In my opinion, this section is not sufficiently elaborated.

i) Page 12, line 262 - secrets? or secretes?

ii) Page 12, line 267-268 - at this stage, why did the authors left out the DPPH-radical scavenging evaluation?

iii) Page 12, line 271-274 - this statement is redundant. Similar meaning has been stated in line 259-263. The authors should provide another explanation along with appropriate references.

iv) Page 13, line 284 - I wonder why at this stage the authors expressed the results of ABTS scavenging activity in Ic50 but in % at the earlier stage of the study?

v) Page 15, line 336-337 - the sentence 'in the case of quinoa fermentation.....' - please provide reference for this statement.

vi) Page 16, line 367-374 - I think that comparison to yogurt fermentation is inappropriate. In my opinion, it is better if the authors compare their results with other studies on cereal or grains fermentation.

5. Conclusions : It is stated in line 384 that the RSM was applied to evaluate the interaction among the three factors. But I could not find the discussion on the interaction of factors anywhere in the manuscript. I can see in Table 3 that the interaction between temperature and agitation speed is significant. The authors should elaborate in the Results and discussion section.

6. Table 4: Are the superscript letters indicating significance of the data are stated correctly? Please check.

Reviewer #2: Reviewer

The research article entitled “Enhanced Antioxidant Activity of Chenopodium formosanum Koidz. by Lactic Acid Bacteria: Optimization of Fermentation Conditions” submitted in your esteemed journal is a good work where the authors have tried to improve the antioxidant activity by lactic acid bacterial and also optimized the fermentation conditions.

Minor Comments: -

What is the method of preparation of bacterial cell suspensions used for fermentation? Also describe about the starter culture for fermentation in material and method section.

In DPPH section what is 1 in the calculation of scavenging activity (%)?

Author cite only one reference for both DPPH and ABTS assay but author used two different calculation method for both DPPH as well as ABTS. Be specific about the citations as well as the calculation.

In section 2.5 author mentioned Wu et al (2020). What is the number of this citation in final list of references???

In section 2.7 line space is different as compared to the other text. Kindly follow the uniformity in whole MS.

The figures are not clearly visible.

At line no 310 author highlights the bracket with red colour. Correct it accordingly.

From section 3.3 it showed that fermentation decrease the carbohydrate, protein and fat content. Kindly explain? What is the reason behind this?

In table 4 correct the word TPC instead of TCP.

Correct the MS with a uniform pattern according to the journal guidelines and also formatting it accordingly.

“After reviewing the manuscript, in my opinion that once the corrections will amended by the author than it will be fit and accepted for publication in your esteemed journal”.

Reviewer #3: The paper describes the utilization of RSM in the optimization of fermentation parameters in order to maximize antioxidant activity. The work is scientifically interesting and sound.

It is good that the authors point out that the increase in antioxidant capacity may be partly related to the enhanced extractability of phenolic compounds. It needs to be remembered that human gut is able to effiicently "extract" and utilize also phenolic compounds that - due to non-extractability in sample preparation - may not be observed when total phenolics/antixoidant capacity is measured.

English needs to be slightly improved.

Some clarification is needed in the following:

1) What is the difference between "Fermented" and "Fermented-RSM" in Table 4?

2) What is "quinoa" mentioned in Table 5? Reference to literature source may be needed. Does "Djulis" denote the nonfermented samples?

6. PLOS authors have the option to publish the peer review history of their article (what does this mean?). If published, this will include your full peer review and any attached files.

Reviewer #1: No

Reviewer #2: **Yes: **DR. Pardeep Kumar

Reviewer #3: No

---

## [Author Response · Author response to Decision Letter 0]

24 Jan 2021

Please see the attached file as our response.

---

## [Editor Report · Decision Letter 1]

27 Jan 2021

PONE-D-20-34408R1

Enhanced Antioxidant Activity of Chenopodium formosanum K oidz. by Lactic Acid Bacteria: Optimization of Fermentation Conditions

PLOS ONE

Dear Dr. Cheng,

Thank you for submitting your manuscript to PLOS ONE. After careful consideration, we feel that it has merit but does not fully meet PLOS ONE’s publication criteria as it currently stands. Therefore, we invite you to submit a revised version of the manuscript that addresses the points raised during the review process.

The general impression is that the authors have put insufficient effort in revising the manuscript. The provided answers to the reviewers' comments are not properly addresses:

R: "And perhaps explain why they selected ABTS and DPPH assays as the anti-oxidant assays for this study."

A: "The reason why we choose ABTS and DPPH assays is because djulis is rich of phenolic compounds."

There are numerous antioxidant assays, which differ in their ability to scavenge particular radicals. The authors have to better explain why they have chosen the stated two. It should be discussed in the manuscript too.

R: "Results and discussion: In my opinion, this section is not sufficiently elaborated."

The authors provided no answer. It is a general remark; however, the Discussion section should be considerably supplemented in several parts where the reviewers' have made their notes.

R: "Conclusions : It is stated in line 384 that the RSM was applied to evaluate the interaction among the three factors. But I could not find the discussion on the interaction of factors anywhere in the manuscript. I can see in Table 3 that the interaction between temperature and agitation speed is significant. The authors should elaborate in the Results and discussion section."

A: "Page 14, lines 323-324: The interaction between temperature and agitation speed can be observed from the results (p < 0.05)."

I cannot accept the provided statement as the elaboration on the concern raised by the reviewer.

A: "We thank the reviewer for the comment, the data presented in % was in comparison to the control based on the equation: DPPH scavenging activity (%) = [1—(A1—A2) / A0] × 100, A0 = DPPH (without samples), A1 = Sample + DPPH, and A2 = Sample (without DPPH). We used this method to screen the optimal LAB for djulis fermentation. When the optimal fermentation protocol was evaluated, IC50 was expressed to provide absolute number to the audience."

This reply should stand in the main text. If two out of three reviewers raised the concern on different presentation of scavenging activities, it is expected readers will also do.

The language must be further polished. Some examples:

L46-47: "...can significantly increase the phenolic compounds," - phenolic compounds cannot be increased. Its amount can instead.

L137 and elsewhere:  "scavenging activity of the djulis" - No article is used when a non-count noun is generic or nonspecific.

L244: "T test" it is rather "t- test"

L363: no contractions such as "it's" are allowed.

Table titles and figure captions: please do not write djulis capitalized.

Figures 3 and 4 should be submitted in the .tiff format. Figure S1 is not a raw image since has been edited.

Significance letters in three histograms representing Figure 1 might be questionable. Please provide statistical working sheet with the raw data as Supporting information.

We look forward to receiving your revised manuscript.

Kind regards,

Branislav T. Šiler, Ph.D.

Academic Editor

PLOS ONE

---

## [Author Response · Author response to Decision Letter 1]

8 Mar 2021

Please see the attached file as our response

---

## [Editor Report · Decision Letter 2]

15 Mar 2021

Enhanced Antioxidant Activity of Chenopodium formosanum K oidz. by Lactic Acid Bacteria: Optimization of Fermentation Conditions

PONE-D-20-34408R2

Dear Dr. Cheng,

We’re pleased to inform you that your manuscript has been judged scientifically suitable for publication and will be formally accepted for publication once it meets all outstanding technical requirements.

Kind regards,

Branislav T. Šiler, Ph.D.

Academic Editor

PLOS ONE
---

## [Editor Report · Acceptance letter]

28 Apr 2021

PONE-D-20-34408R2 

Enhanced Antioxidant Activity of *Chenopodium formosanum* Koidz. by Lactic Acid Bacteria: Optimization of Fermentation Conditions 

Dear Dr. Cheng:

I'm pleased to inform you that your manuscript has been deemed suitable for publication in PLOS ONE. Congratulations! Your manuscript is now with our production department. 

Kind regards, 

on behalf of

Dr. Branislav T. Šiler 

Academic Editor

PLOS ONE